# Assessment of Hospital Readiness to Respond to COVID-19 Pandemic in Jordan—A Cross Sectional Study

**DOI:** 10.3390/ijerph20031798

**Published:** 2023-01-18

**Authors:** Eman Zmaily Dahmash, Thaira Madi, Ahmad Shatat, Yazan Oroud, Samar Khaled Hassan, Omaima Nassar, Affiong Iyire

**Affiliations:** 1Faculty of Health, Science, Social Care and Education, School of Life Sciences, Pharmacy and Chemistry, Kingston University, London KT1 2EE, UK; 2Department of Accreditation, Healthcare Accreditation Council, Amman 11181, Jordan; 3Department of Accounting, Faculty of Business, Isra University, Amman 11622, Jordan; 4Aston Pharmacy School, Aston University, Birmingham B4 7ET, UK

**Keywords:** COVID-19, emergency preparedness, pandemic, Jordan, WHO hospital readiness checklist

## Abstract

During the global COVID-19 pandemic, hospitals faced tremendous pressure to cope with the emergency preparedness situations needed to cater for the influx of patients while maintaining their essential services. This study aimed to assess the level of readiness of hospitals in Jordan to respond to the COVID-19 pandemic using the WHO hospital readiness checklist. A cross-sectional survey using the modified and validated checklist was conducted in Jordan between 15 May and 15 June 2021. The checklist entailed ten key response functions with a total of 60 activities. Data from 22 hospitals were collected through a structured survey process by two surveyors for each hospital. The overall readiness score of hospitals was 1.77 ± 0.20, with a lower overall score in the northern region (1.65 ± 0.24) than the middle (1.86 ± 0.07) or southern (1.84 ± 0.14) regions. The diagnosis response function scored highest (1.95); but despite efforts, contingency plan development was not met by most hospitals, with a total score ≤ 1.45. Provision of psychological support and occupational health support to ensure the wellbeing of staff scored below average. Outcomes from this survey exposed gaps while offering a framework for upcoming endeavors to improve hospital readiness for any potential pandemic.

## 1. Introduction

The COVID-19 pandemic has created significant clinical and operational challenges for hospital leaders across the globe, especially when COVID-19 cases have increased beyond a hospital’s capacity to manage them [1,2,3,4]. Thus, international experience has validated the importance of adequate preparedness and readiness in hospitals to achieve effective crisis management [4,5]. The United States Agency for International Development (USAID) Health Service Delivery supported the Ministry of Health (MoH) in Jordan to adapt the World Health Organization (WHO)—East Mediterranean Regional Office (EMRO) Hospital Readiness Checklist for COVID-19 to the Jordanian healthcare system [6]. This checklist aims to support the preparedness of hospitals in Jordan to effectively manage increases in COVID-19 cases, including planning for any such unexpected future incidents that would increase surge capacity. The checklist also ensures the continuity of all other essential services, while establishing a safe environment for health care professionals (HCPs) and others visiting the hospital [7,8,9].

During the COVID-19 pandemic, hospitals were the backbone for effective response, and their resilience is critical to achieving a balanced health coverage not only for COVID-19-affected individuals but for persons with other illnesses [1,5,10]. Multifaceted collaborations along with multi-level policies are therefore mandatory to strengthen hospitals’ resilience and to prepare the healthcare system for upcoming pandemics [11]. The COVID-19 pandemic transformed the vision towards economy, and society and governments are now required to implement strategies that produce more inclusive and resilient future interventions based on lessons learnt from the COVID-19 pandemic [4,5]. 

Healthcare experts globally have emphasized the importance of strengthening health systems, particularly hospitals, for effective pandemic response to protect both patients’ rights to health and staff [12,13]. Throughout the Eastern Mediterranean region, poor contingency planning and lack of supplies threaten hospitals’ readiness to respond to pandemics [5,14,15]. Overall, hospitals within this region normally run at their maximum capacity normally, and hence a slight increase in patient numbers during a pandemic can put a large strain on hospitals’ capacities [5,16]. The Eastern Mediterranean healthcare systems face multifaceted challenges and hence, there is a need for periodic assessment of hospitals’ readiness to deal with pandemics. That will enable policymakers to take informed decisions that will strengthen hospitals’ readiness for various types of pandemics and create a resilient healthcare system [15]. 

In Jordan, the COVID-19 cases showed a peak that spanned between the first and second quarter of 2021, with registered daily cases initially reported in hundreds and later in thousands [17,18]. This pandemic wave has posed a significant burden on the healthcare system with high isolation room occupancy rates, and very high demand for ventilators and intensive care unit (ICU) beds, particularly in the northern and middle regions of Jordan. At the beginning of the second quarter of 2021, the occupancy rate of ICU beds for COVID-19 cases exceeded 73% in the northern region and 77% in the middle region in Jordan [19]. An alarming 50% occupancy rate of ventilators was also reported in various regions [17].

The healthcare accreditation council (HCAC), the national independent accrediting body in Jordan, has utilized the readiness checklist that fits the Jordanian context, to support accredited hospitals in coping with the pandemic. The readiness checklist is based on the World Health Organization’s Regional Office for the Eastern Mediterranean Hospital Readiness Checklist for COVID-19, adopted by USAID Health Service Delivery (HSD) project in collaboration with its counterparts at the MoH Communicable Disease Directorate (CDD). This checklist consists of ten response functions that include 60 activities. Designed to identify the hospital’s strengths, weaknesses, and gaps related to COVID-19 preparedness [6,7]. Therefore, the aim of this study was to determine the level of readiness of accredited hospitals in Jordan to respond to the COVID-19 pandemic. The outcome of this work will provide knowledge to guide the development of a roadmap and create strategies to address priority areas and improve preparedness for a sustained pandemic response.

## 2. Methods

### 2.1. Research Design and Sampling Strategy

This was a cross-sectional study among accredited hospitals in Jordan. The process was based on a one-day survey to the targeted hospitals during the period between 20 of May and 7 of June 2021 by certified surveyors. The surveyors received training on the survey tool (WHO-Hospital Readiness Checklist for COVID-19). Accredited hospitals were invited to participate in the study and a total of 22 hospitals were included. Inclusion criteria included accredited public, private or university hospitals in Jordan. Exclusion criteria included non-accredited hospitals and military hospitals (due to complexity of the approval process). Only accredited hospitals were selected because the HCAC wanted to explore how HCAC accreditation requirements helped hospitals by raising their readiness to deal with pandemics such as COVID-19. Further, assessing their readiness and providing them with identified gaps and areas for improvement was part of HCAC support to accredited hospitals.

In early 2021, the checklist was communicated to all public hospitals by the USAID and the Ministry of Health in Jordan. The hospitals’ managers were asked to use the tool to assess their readiness level and oversee the completion process of the survey. One month prior to the assessment, HCAC shared the checklist with all participating hospitals and sent an official letter to hospitals to inform them about the assessment date and the assessment team.

### 2.2. Survey Tool

The WHO checklist for hospital readiness for COVID-19 was amended in four ways to transform it into a survey instrument. First, some elements on the checklist were tailored or further explained to enhance clarity. Second, some elements on the checklist were converted to a single answer for questions that contained two or more elements. These items were integrated if the sub parts were closely related and where a single response was sufficient; or separated into two questions. Third, some questions were removed as they were part of other domains or were not relevant to the hospital settings in Jordan. Fourth, the verification of each question was set at three levels: met, partially met or not met. The tool was validated by experts from the United States Agency for International Development (USAID) Health Service Delivery, and further updates were made [20]. The final validated tool was granted from the USAID Health Service Delivery for use by hospitals [21]. HCAC did not participate in the validation of the tool; they used the final validated survey tool. The final survey tool contained, as with the original WHO checklist, ten responses with a total of 60 questions. The responses consisted of the following areas: leadership and coordination; operational support; logistics and supply management; information; communication; human resources; continuity of essential services and surge capacity; rapid identification; diagnosis; isolation and case management; and infection prevention and control (Appendix A). 

### 2.3. Scoring

The 60 activities were framed into questions with three (met, partially met, or not met) response options to be assessed by the surveyor. The “met” score was given 2 marks in the analysis, the “partially met” was given 1 mark and the “not met” was scored as zero. Therefore, the higher the score, the higher the readiness level of the hospital for dealing with COVID-19 cases. 

### 2.4. The Survey Process and Data Collection

The survey process was based on two healthcare professional surveyors per hospital. Conflict of interest was addressed based on HCAC criteria. Twenty-four surveyors received training on how to conduct the survey. All surveyors were HCAC-certified surveyors. The HCAC follows a well-defined system for assigning survey teams that are competent for the task, and that have no conflict of interest and no known cultural, religious, or social issues that would affect the success of the assignment. (All surveyors are required to declare No Conflict of Interest at the beginning of their relationship with HCAC and continuously for each contract renewal (every two years) based on HCAC Conflict of Interest Policy and Procedures.) The assigned surveyors for each survey must assure that no conflict of interest exists for conducting that survey. Examples of preventive measures that HCAC has followed to prevent the conflict of interest are: surveyors cannot do surveys for the same organization more than one time within a three-year period; furthermore, they cannot do surveys of the organizations they have been employed by until after three years of leaving them. HCAC manages the survey process using an automated system.

The data collected from the application of the checklist was used to assess the preparedness of the hospitals to effectively deal with COVID-19 cases. Use of the checklist also ensures the continuity of other essential services and establishes a safe working environment for healthcare providers and patients. 

Qualitative and quantitative data to ascertain the status of compliance to each activity and the associated response function were collected for this work using three sources including document review, structured interviews and observations. Interviewees included the hospital director, the hospital management team and healthcare providers. Observations included observing the practice, processes and environment. To avoid biases in data collection, a detailed assessment guide for each activity was made to support the assessment process and unify the work (Appendix A). The assessment guide mentioned clearly for each of the 60 activities within the tool the survey process/method that the surveyors had to use to assess the activity completely. These survey process/methods included (as applicable) what types of documents the surveyors needed to review, who needed to interview staff members and what questions needed to be asked, and what observations the surveyors had to make during their tours and visits. In addition, the teams used a unified agenda that spanned over 6 h (Appendix A). The surveyors’ training included a training on the assessment guide. In addition to that, each survey was conducted by a team of two surveyors, and there was a quality assurance process for the survey report. All study tools (assessment guide Appendix A, list of required documents Appendix A) were reviewed by the USAID Health Service Delivery experts, piloted, and modified accordingly. HCAC and Ethical considerations were applied during the surveys. Hospitals’ participation was voluntary, and participants provided their informed verbal and written consent after receiving all relevant information introducing the project. The results of each survey were reported by the two surveyors and findings were shared with the hospitals to enable them to address all gaps.

### 2.5. Ethical Approval

The study received ethical approval from the Scientific Research Ethics Committee at Isra University (approval number SREC/221031031). All surveyors signed a confidentiality/non-disclosure agreement.

### 2.6. Statistical Data Analysis

The analysis was based on the ten key responses of the WHO’s Hospitals Readiness Checklist as the guiding framework. The results of the 22 relevant and complete reports were reviewed and a simple descriptive analysis to describe participating hospitals’ demographic characteristics was conducted using Microsoft Excel. Continuous data were presented as mean ± standard deviation (SD) for normally distributed variables. Categorial data were presented as percentages and frequencies. The independent two-sample *t*-test was used for the analysis of two samples and one-way ANOVA was used for multiple factors. Multiple regression analysis was used to estimate correlation between various responses. A *p* value of <0.05 was considered as statistically significant. The statistical analyses were carried out using SPSS (version 25, IBM, Armonk, NY, USA).

## 3. Results

### 3.1. Participating Hospital Characteristics

A total of 22 hospitals participated in the study (public = 15, private = 6, university = 1). The university hospital was added to public hospitals as it is a publicly funded hospital. Table 1 details the initial characteristics of the participating hospitals. The majority of participants (*n* = 11, 50.0%) were public hospitals within the middle region and have been accredited for over 5 years (*n* = 14, 63.6%). 

### 3.2. Hospitals’ Readiness According to Response Function

The overall compliance score among all study groups for all the response functions exceeded 1.77 (88.5%). However, the lowest was for public hospitals with a total score of 1.72 (86%). Diagnosis response function scored the highest among all groups (1.95, [97.5%]), followed by information and communication (1.94, [97%]). Table 2 details the overall compliance rate among participants stratified by response function and hospital type. However, there was a statistically significant difference among hospitals according to hospital type. Private hospitals’ overall scores were higher (*p*-value 0.009). The main variations among hospitals according to hospital type was pertinent to operational support, logistics and supply management (*p*-value 0.027), as well as human resources (*p*-value 0.008). 

When compliance score was stratified according to hospitals’ location (north, middle, or south), the north region reported lower overall score (1.65). However, there was no statistically significant difference among regions (see Table 3). 

### 3.3. Readiness According to Detailed Response Functions

Table 4 presents the readiness scores for the first two response functions (leadership and coordination; operational support; logistics and supply management) among the entire study group. The lowest score found in the leadership and coordination response function was pertinent to the development of a contingency plan that had an overall score of 1.45 ± 0.80; however, public hospitals, hospitals in the south region, and those with accreditation status less than 5 years scored better. Despite variations among hospitals, the difference was not statistically significant. Furthermore, formulating a post-mortem contingency plan scored the lowest among hospitals (1.27 ± 0.94). In this domain, private hospitals’ performance was significantly outstanding (2.0 ± 0.0), while public hospitals scored low (1 ± 0.97) (*p*-value 0.021). This could be justified due to the increase in numbers of deaths that is beyond the capacity of the public hospitals, particularly in peripheral hospitals. Coordination with senior management, identifying security constraints and optimizing security measures scored high among all hospitals. 

Information, communication, and human resources scores are reported in Table 5. The compliance score of the hospitals in each question ranged from 1.86 ± 0.35 to 2.0 in the information and communication domain, while from 0.82 ± 0.96 to 1.95 ± 0.21 for human resources. The availability of multidisciplinary psychosocial support teams was not prevalent in all southern hospitals and most public hospitals. A statistically significant difference between public and private hospitals was evident (*p*-value 0.001). Variations among hospitals were significant in terms of informing and training staff who were planned to be reallocated in accordance with their anticipated roles and responsibilities. In this regard, north-located hospitals were not provided with adequate training when compared to the middle or southern region hospitals (*p*-value 0.019). A significant variation was observed in the score for using occupational health mechanisms that ensure the wellbeing and safety of personnel between public and private hospitals. Private hospitals were able to embrace mechanisms that ensured the wellbeing of personnel during the response and to monitor stress-related impact on staff due to extended working hours (*p*-value 0.008).

The readiness scores for all hospitals in the next four functions (surge capacity, continuity of essential services, rapid identification and diagnosis) are depicted in Table 6. Apparently, these are critical processes within hospital readiness. Overall, the diagnosis response function scored high in all its subcategories, while the lowest was related to developing a system for alternative triage, particularly in public hospitals (0.88 ± 1.03). 

Table 7 compares the scores between hospitals stratified according to their location, type and accreditation years in the last two response functions (isolation and case management; infection prevention and control). The overall score for isolation and case management was below average. The lowest was for the function that required patients to be placed in an adequately ventilated single room (≥12 air changes/hour and when single rooms are not available, patients suspected of having COVID-19 should be grouped together). In this function, the public hospitals’ score was significantly lower than that of the private hospital (*p*-value 0.036). However, southern, and middle region hospitals’ readiness levels were better than that of the north-located hospitals in ensuring strict supervision on the implementation of infection prevention and control measures, despite a low score for all. 

### 3.4. Regression Analysis

Using multiple linear regression, it was found that continuity of essential services was the only response factor that could be predicted. The results revealed that operational support and human resources were more likely to enhance the continuity of essential services during the COVID-19 pandemic (Table 8). Proper human resource management provided high level of correlation to and prediction of effective compliance with continuity of essential services.

## 4. Discussion

The primary aim of the WHO COVID-19 hospital readiness checklist was to support hospitals to assess their current situation to manage the emergency, identify gaps in services necessary to respond to the COVID-19 pandemic and build strategies to enhance hospital readiness [6]. Therefore, the aim of this research was to use the checklist to evaluate hospital readiness levels and identify aspects in practice that need to be addressed by policy makers and senior hospital leadership and to ensure a synchronized approach to improvement, which may benefit post-COVID-19 management and also combating any future emergencies.

The study was conducted almost one year after the first case was reported in Jordan, hence not at the early stages of the pandemic. Yet, the findings of this study showed gaps in various aspects, particularly in leadership, planning, human resources and key processes (e.g., rapid identification, continuity of essential services and surge capacity). The COVID-19 pandemic has revealed disparities in health systems and consequently the limited access to several healthcare services in the Eastern Mediterranean Region [11]. The low score in the leadership domain could be justified as hospital leaders were exposed to this situation (dealing with and managing a pandemic) for the first time; they faced new challenges that required them to perform unfamiliar tasks, which may have been outside the scope of their previous practices and experience. A set of new managerial competencies defined for healthcare leaders is needed to address the pandemic, which may be better provided and supported by competency-based training courses [22].

Effective response to pandemics is driven by collaboration and coordination among various systems, starting from engaged leadership to supportive human resources [23,24,25]. Although private hospitals’ readiness score was better than that of the public hospitals, they shared poor scoring when it pertained to the development of contingency plans. The lowest score was for private hospitals (1.17 ± 0.98). Lack of contingency planning could partly explain the moderate level of readiness scores by hospitals in this study.

The findings of this study also revealed variations in the level of readiness according to location; hospitals located in the northern region of Jordan demonstrated the lowest scores in most domains (despite lack of significant difference, which could be justified by the uneven distribution of the number of hospitals). In comparison with the other two regions, during November 2021, hospitals within the northern region experienced higher occupancy rates, particularly in intensive care units (ICUs) (42%), while the middle and southern ICU occupancy rates reached 40% and 14%, respectively. Utilization of ventilators in the northern hospitals was 22% while the middle and south did not exceed 13% [26]. The overall pressure on southern hospitals was lower, which justified the higher readiness score. Furthermore, in the north, the main hospital that was concerned with COVID-19 patients was King Abdullah University Hospital (KAUH), which was transformed into a mainly COVID-19 cases site, with drastic reduction in the provision of other essential services to cope with the high demand of COVID-19 patients [27]. Therefore, this justified the low score in continuity of essential services (1.46 ± 0.8). Further, in the northern region they had more COVID-19 cases and went into full lockdown more than once. Actually, the increase in cases began in the northern region. Research by Al-Qudah et al. [27] reported a decline in overall routine services provided to patients in KAUH (the main hospital within the northern region), suggesting a potential increase in other disease complications, particularly among low-income families. Allocated resources for healthcare are generally scarce; thus, government and healthcare leaders need to identify strategies for fair allocation of available resources to enable the continuum of healthcare for pandemic and routine cases alike [28].

The findings of the WHO second pulse survey for the continuity of essential health services after one year of the pandemic for 135 countries are in concordance with the findings of this study. The results revealed that considerable disruptions of essential services continued in almost 90% of countries. Furthermore, 94% reported disruption of at least one service, while 34% reported a disruption in over 50% of their services. Only 6% of the participating countries reported that services were not disrupted [29].

In Jordan, a total of 106 hospitals are providing secondary and tertiary care using 12,081 hospital beds, of which public hospitals account for 67% [27]. During the pandemic, the public and private healthcare sectors partnered to strengthen their combating the pandemic [30,31,32] through the National Centre for Security and Crisis Management [26,27]. However, in the face of the pandemic, the private hospital’s overall readiness score was significantly higher than that of the public hospitals (*p*-value 0.009). The key domains that demonstrated significantly better scores were those related to human resources, operational support, and logistics and supply management.

Operational support and logistics and supply management scores among hospitals varied. However, our study showed that the pandemic instigated improvement in this domain particularly in maintenance, security, ambulance and transport, as well as backup arrangements. Nevertheless, the key challenges were pertinent to provision of post-mortem care contingency plans, which scored 1.27 ± 0.94, followed by managing work teams, followed by identifying additional storage facilities. This could be justified due to the increase in numbers of deaths that was beyond the capacity of the public hospitals, particularly in peripheral hospitals. Similar results were reported by Ogoina et al. in Nigeria [33], where only 30% of the hospitals developed a post-mortem contingency plan and facility. Despite clear guidelines [28,29], public hospitals did not prepare such aspects. Similar findings were reported where allocating additional storage capacity was a key constraint in supply chain management during COVID-19 [34,35,36]. The sudden increase in demand was a reason behind this.

As for ensuring a procedure for the management of work teams, including rest areas, safe transportation and staff wellbeing was a key challenge during the pandemic. Frontline staff were subject to unprecedented strain during the pandemic that stemmed from high workload, fear of infection and isolation [37,38]. Several wellbeing strategies were reported, such as provision of rest areas, increasing staffing numbers, easy access to proper meals, and personal protective equipment (PPE) as valuable factors for the wellbeing of healthcare professionals—particularly frontliners [37]. Apparently, private hospitals in Jordan were able to provide such measures while public hospitals were not; this could be attributed to the high workload in public hospitals when compared to private.

Two key elements in human resources were suboptimal in this study. The first was related to the provision of multidisciplinary psychosocial support and the second was related to the use of occupational health mechanisms that ensure wellbeing and staff safety. These two domains were significantly evident in public hospitals, and as discussed earlier, workload and limited resources in public hospitals limited the provision of such highly needed support. Staff wellbeing was also reported as a challenge in another study where only five out of the 20 hospitals surveyed in Nigeria provided accommodation for frontline staff and only 20% of hospitals had multidisciplinary psychosocial support [33]. Hospitals need to develop and implement a mitigation plan towards the negative psychological and occupational concerns of frontline staff during pandemics. Several studies have reported effective strategies for psychological support and wellbeing for frontliners which include: clear communication [39], safety and provision of PPEs [40], uninterrupted safe rest areas [41], working a maximum of four consecutive shifts followed by at least three days-off [42], provision of training and education [39], peer support [43] and wellbeing drop in sessions [40].

Surge capacity is fundamental in emergency preparedness. Research reported that it entails three core elements; staff, stuff and system [44,45]. These elements were part of the surge capacity response function in this study, which was reported to be low (1.55 ± 0.86). This is related to calculation of the maximum admission capacity score (1.64 ± 0.73) and ability to convert regular rooms into isolation rooms score (1.64 ± 0.73). In this study, private hospitals’ surge capacity response was high when compared to public hospitals. This could be attributed to lower occupancy rate in private hospitals during the pandemic when compared to public hospitals. Research findings have also reported a relationship between surge capacity and occupancy rate; high bed occupancy rate resulted in low unoccupied beds, therefore limited surge capacity [44,46].

During a pandemic, an effective hospital emergency response should be able to develop rapid identification processes that enable effective triage systems and alternative triage systems, which will reduce mortality and morbidity [47,48]. Although private hospitals had adequate triaging systems, all hospitals in our study did not develop alternative triaging systems (overall score 1.09 ± 1.02). A study by Taylor (2022) reported the benefits of tele-triaging systems which emerged as an opportunity to address serious challenges during the contagious COVID-19 pandemic. This would reduce the risk of viral transmission, reduce the use of PPE and enable the use of self-quarantine staff to pursue some duties [49].

Our study findings also showed that hospitals demonstrated high level of readiness in various aspects related to diagnosis, isolation, case management and infection prevention and control. The highest readiness scores were reported for diagnosis response (1.95 ± 0.21). The reasons for these high scores could possibly be attributed to the excellent communication of all relevant updates and information related to the pandemic that scored high as well (1.94). Furthermore, it was proposed that all hospitals prioritize implementation of actions under infection prevention and control responses, as they were associated with detection and prevention of COVID-19 infection among workers and patients. However, strict supervision of the implementation of infection prevention and control measures was inadequate among all hospitals (1.55 ± 0.74).

Interestingly, our results showed that continuity of essential services tend to be positively impacted by the proper implementation of operational support, logistics, supply management and human resources (HR) responses (*p* value 0.001 and 0.000, respectively). Bennett (2021) reported that the pandemic has enhanced global cooperation, strengthened private-public partnerships, and shaped the future of human resource management [50]. The results are a call for human resource management as the functions related to HR positively impact services. Several studies highlighted the importance of HR in crisis management and developed various frameworks to enable better management of HR during pandemics [50,51,52].

### Strengths and Limitations

To the best of our knowledge, this is the first study in the Middle East that investigated hospital readiness according to the WHO checklist (Hospital Readiness Checklist for COVID-19). The use of a validated assessment tool is another strength of this study. Furthermore, the survey process enabled the collection of data using more than one source (document review, interviews, and observations) as well as employing two surveyors for each hospital, which reduced the bias.

However, there are some limitations to this study. First, there was a limited number of studies that assessed hospital readiness using the checklist during the same period of the COVID-19 pandemic in the Middle East in particular, which limited the ability to compare the findings of this study with similar countries’ facilities. Second, due to the study design utilizing only accredited hospitals, it was difficult to generalize the findings to cover all hospitals; however, it is also important to highlight that this was clearly stated in the inclusion/exclusion criteria. Finally, all participating private hospitals are in the middle region in Jordan, which prevents the generalizability of private hospitals-related data.

## 5. Conclusions

The main objective of the study was to assess the level of hospitals’ readiness to deal with the COVID-19 pandemic in Jordan. The hospitals’ general preparedness level for the management of COVID-19 pandemic was moderate. The findings of this survey revealed aspects in practice that require strengthening. There is much to be learned to enhance emergency preparedness activities in healthcare systems. There is also a need to develop emergency management frameworks based on the lessons learnt form the COVID-19 pandemic. It is vital that we address the challenges, gaps and missed opportunities of the previous pandemic if we are to prevent replicating them in future emergencies. The Hospital Readiness Checklist findings will assist hospital leaders in making informed decisions, developing a roadmap, and creating strategies to respond to COVID-19 and other emergency cases in their hospitals.

## Figures and Tables

**Table 1 ijerph-20-01798-t001:** Participating hospitals’ characteristics.

Demographics		Overall *n* = 22 (%)	Public Hospitals *n* = 15 (%)	Private Hospital *n* = 6 (%)	University Hospital *n* = 1 (%)
**Location**	North	8 (36.4)	7 (46.7)	0 (0)	1 (100)
	Middle	11 (50)	5 (33.3)	6 (100)	0 (0)
	South	3 (13.6)	3 (20)	0 (0)	0 (0)
**Accreditation**	<5 years	8 (36.4)	5 (33.3)	2 (33.3)	1 (100)
	>5 years	14 (63.6)	10 (66.7)	4 (66.7)	0 (0)

**Table 2 ijerph-20-01798-t002:** Average compliance score of participating hospitals according to response function, stratified according to hospital type.

Response Function Compliance Score	Overall (*n* = 22)	Public Hospitals ‡ (*n* = 16)	Private Hospital (*n* = 6)	*p*-Value
1. Leadership and Coordination	1.72 ± 0.42	1.70 ± 0.47	1.77 ± 0.27	0.737
2. Operational Support, Logistics and Supply Management	1.76 ± 0.26	1.69 ± 0.27	1.96 ± 0.06	0.027 *
3. Information and Communication	1.94 ± 0.16	1.94 ± 0.17	1.93 ± 0.10	0.894
4. Human Resources	1.69 ± 0.33	1.58 ± 0.32	1.97 ± 0.05	0.008 **
5. Surge Capacity	1.73 ± 0.47	1.65 ± 0.53	1.93 ± 0.16	0.224
6. Continuity of Essential Services	1.73 ± 0.54	1.65 ± 0.61	1.94 ± 0.14	0.269
7. Rapid Identification	1.58 ± 0.47	1.46 ± 0.48	1.89 ± 0.27	0.053
8. Diagnosis	1.95 ± 0.21	1.94 ± 0.25	2.0 ± 0.0	0.569
9. Isolation and Case Management	1.73 ± 0.23	1.7 ± 0.23	1.83 ± 0.23	0.252
10. Infection Prevention and Control	1.88 ± 0.19	1.87 ± 0.21	1.90 ± 0.17	0.758
**Overall**	**1.77 ± 0.20**	**1.72 ± 0.16**	**1.92 ± 0.08**	**0.009 ****

‡ University hospital was added to the public hospitals, * Level of significance < 0.05, ** level of significance < 0.01.

**Table 3 ijerph-20-01798-t003:** Average compliance score of participating hospitals according to response function stratified according to hospital location.

Response Function Compliance Score	North	South	Middle	*p*-Value
	(*n* = 8)	(*n* = 3)	(*n* = 11)	
1. Leadership and Coordination	1.5 ± 0.6	1.93 ± 0.12	1.82 ± 0.23	0.171
2. Operational Support, Logistics and Supply Management	1.67 ± 0.32	1.74 ± 0.23	1.84 ± 0.22	0.391
3. Information and Communication	1.95 ± 0.14	2 ± 0	1.91 ± 0.19	0.672
4. Human Resources	1.48 ± 0.41	1.73 ± 0.09	1.83 ± 0.22	0.0612
5. Surge Capacity	1.5 ± 0.68	2 ± 0	1.82 ± 0.28	0.205
6. Continuity of Essential Services	1.46 ± 0.8	2 ± 0	1.85 ± 0.27	0.204
7. Rapid Identification	1.67 ± 0.36	1.67 ± 0.58	1.48 ± 0.54	0.663
8. Diagnosis	1.88 ± 0.35	2 ± 0	2 ± 0	0.459
9. Isolation and Case Management	1.66 ± 0.24	1.86 ± 0.14	1.75 ± 0.24	0.432
10. Infection Prevention and Control	1.8 ± 0.25	1.88 ± 0.22	1.93 ± 0.13	0.367
** Overall **	**1.65 ± 0.24**	**1.86 ± 0.07**	**1.84 ± 0.14**	** 0.073 **

University hospital was added to the public hospitals.

**Table 4 ijerph-20-01798-t004:** Detailed average compliance score of participating hospitals according to response function stratified according to hospital location, type and years of accreditation for response functions (leadership and coordination as well as operational support and logistics and supply management).

Readiness Response	Overall	Hospital Location				Hospital Type			Years of Accreditation		
	Total Score (*n* = 22)	Middle (*n* = 11)	North (*n* = 8)	South (*n* = 3)	*p*-Value	Public (16)	Private (6)	*p*-Value	<5 Years (*n* = 9)	≥5 Years (*n* = 13)	*p*-Value
**Leadership and Coordination**											
Establish/activate Hospital Incident Management team involving representatives from all departments and units.	1.59 ± 0.59	1.73 ± 0.47	1.38 ± 0.74	1.67 ± 0.58	0.447	1.56 ± 0.63	1.67 ± 0.52	0.722	1.56 ± 0.53	1.62 ± 0.65	0.822
Designate a secure, easily accessible, and well-equipped Hospital Emergency Operations Centre.	1.86 ± 0.47	2 ± 0	1.63 ± 0.74	2 ± 0	0.2	1.81 ± 0.54	2 ± 0	0.415	2 ± 0	1.77 ± 0.60	0.265
Assign roles and responsibilities for the different response functions.	1.86 ± 0.47	2 ± 0	1.63 ± 0.74	2 ± 0	0.2	1.81 ± 0.54	2 ± 0	0.415	2 ± 0	1.77 ± 0.60	0.265
Develop contingency plans for staffing, logistics, budget, procurement, security and treatment.	1.45 ± 0.80	1.36 ± 0.81	1.38 ± 0.92	2 ± 0	0.468	1.56 ± 0.73	1.17 ± 0.98	0.313	1.56 ± 0.73	1.38 ± 0.87	0.634
Compile an up-to-date directory of telephone numbers, residences and email addresses of staff and their representatives.	1.82 ± 0.588	2 ± 0	1.5 ± 0.93	2 ± 0	0.161	1.75 ± 0.68	2 ± 0	0.388	2 ± 0	1.69 ± 0.75	0.237
**Operational Support, Logistics and Supply Management**											
Coordinate with administrative board of hospital to ensure the continuous provision of essential medications and supplies.	2 ± 0	2 ± 0	2 ± 0	2 ± 0		2 ± 0	2 ± 0		2 ± 0	2 ± 0	
Estimate consumption of essential supplies and pharmaceuticals to ensure the continuous provision of essential medications and supplies.	1.77 ± 0.53	1.91 ± 0.30	1.63 ± 0.74	1.67 ± 0.58	0.5	1.69± 0.60	2 ± 0	0.225	1.78 ± 0.67	1.77 ± 0.44	0.971
Identify storage facilities for additional stock that meet the storage demands with respect to temperature, humidity, and cold chain.	1.59 ± 0.67	1.64 ± 0.67	1.38 ± 0.74	2 ± 0	0.382	1.56 ± 0.73	1.67 ± 0.52	0.753	1.78 ± 0.44	1.46 ± 0.78	0.284
Ensure a procedure for the management of work teams.	1.5 ± 0.80	1.55 ± 0.82	1.5 ± 0.76	1.33 ± 1.16	0.928	1.31 ± 0.87	2 ± 0	0.072	1.44 ± 0.88	1.54 ± 0.78	0.794
Ensure a mechanism for the prompt maintenance and repair of all equipment required for essential services.	1.91 ± 0.43	2 ± 0	1.75 ± 0.71	2 ± 0	0.438	1.88 ± 0.5	2 ± 0	0.553	2 ± 0	1.85 ± 0.56	0.419
Ensure a procedure for managing ambulances for transportation between hospitals and for the inventory of available vehicles, and a procedure to protect ambulance crew and disinfect ambulance vehicles and equipment after each use.	1.91 ± 0.29	2 ± 0	1.88 ± 0.35	1.67 ± 0.58	0.209	1.88 ± 0.34	2 ± 0	0.388	2 ± 0	1.85 ± 0.38	0.237
Ensure the availability of appropriate back-up arrangements for essential lifelines, including water, electric power, and oxygen.	1.91 ± 0.29	1.91 ± 0.30	2 ± 0	1.67 ± 0.58	0.257	1.88 ± 0.34	2 ± 0	0.388	2 ± 0	1.85 ± 0.38	0.237
Solicit the input of hospital security in identifying potential security constraints and optimizing the control of facility access, essential pharmaceutical stocks, patient flow, traffic and parking; seek support from local security forces to augment hospital security, if needed.	2 ± 0	2 ± 0	2 ± 0	2 ± 0		2 ± 0	2 ± 0		2 ± 0	2 ± 0	
Formulate a postmortem care contingency plan with appropriate partners, for managing an increased need for postmortem care and disposition of deceased patients, and guidelines for the disposal and transport of corpses resulting from the emergency.	1.27 ± 0.94	1.55 ± 0.82	0.88 ± 0.991	1.33 ± 1.155	0.316	1 ± 0.97	2 ± 0	0.021 *	1.44 ± 0.88	1.15 ± 0.99	0.487

* Level of significance *p* < 0.05.

**Table 5 ijerph-20-01798-t005:** Detailed average compliance score of participating hospitals according to response function stratified according to hospital location, type, and years of accreditation for response functions (information and communication, and human resources).

Readiness Response	Overall	Hospital Location				Hospital Type			Years of Accreditation		
	Total Score (*n* = 22)	Middle (*n* = 11)	North (*n* = 8)	South (*n* = 3)	*p*-Value	Public (16)	Private (6)	*p*-Value	<5 Years (*n* = 9)	≥5 Years (*n* = 13)	*p*-Value
**Information and Communication**											
Establish procedures and assign personnel to collect, confirm and validate data and information related to the emergency.	1.91 ± 0.30	1.91 ± 0.30	1.88 ± 0.35	2 ± 0	0.835	1.94 ± 0.25	1.83 ± 0.41	0.473	1.89 ± 0.33	1.92 ± 0.28	0.796
Provide a standardized form for internal reporting.	2 ± 0	2 ± 0	2 ± 0	2 ± 0		2 ± 0	2 ± 0		2 ± 0	2 ± 0	
Communicate regularly with staff and stakeholders about their roles and responsibilities in managing the COVID-19 crisis, clinical triage, patient prioritization and management, hospital epidemiology, reporting requirements and security measures.	2 ± 0	2 ± 0	2 ± 0	2 ± 0		2 ± 0	2 ± 0		2 ± 0	2 ± 0	
Ensure that all internal protocols, communication lines and standard operating procedures are easily accessible.	1.86 ± 0.35	1.82 ± 0.41	1.88 ± 0.35	2 ± 0	0.744	1.88 ± 0.34	1.83 ± 0.41	0.811	1.78 ± 0.44	1.92 ± 0.28	0.353
Ensure reliable and sustainable primary and back-up communication systems.	1.91 ± 0.43	1.82 ± 0.60	2 ± 0	2 ± 0	0.629	1.88 ± 0.5	2 ± 0	0.553	1.78 ± 0.67	2 ± 0	0.238
**Human Resources**											
Adapt human resource management to ensure adequate staff capacity and continuity of operations.	1.91 ± 0.43	2 ± 0	1.75 ± 0.71	2 ± 0	0.438	1.88 ± 0.5	2 ± 0	0.533	2 ± 0	1.85 ± 0.56	0.419
Prioritize staffing needs by unit or service and distribute personnel accordingly.	1.91 ± 0.43	2 ± 0	1.75 ± 0.71	2 ± 0	0.438	1.88 ± 0.5	2 ± 0	0.533	2 ± 0	1.85 ± 0.56	0.419
Communicate staffing needs for transmission scenarios to the hospital administrative board.	1.86 ± 0.47	1.91 ± 0.30	1.75 ± 0.71	2 ± 0	0.681	1.81 ± 0.54	2 ± 0	0.415	2 ± 0	1.77 ± 0.6	0.265
Estimate staff absenteeism in advance and monitor it continuously.	1.64 ± 0.66	1.73 ± 0.65	1.5 ± 0.76	1.67 ± 0.58	0.774	1.5 ± 0.73	2 ± 0	0.114	1.56 ± 0.73	1.69 ± 0.63	0.643
Apply policies and procedures for screening and work restrictions for exposed or ill healthcare personnel	1.95 ± 0.21	2 ± 0	1.88 ± 0.35	2 ± 0	0.438	1.94 ± 0.25	2 ± 0	0.553	2 ± 0	1.92 ± 0.28	0.419
Inform and train staff who are planned to be reallocated in accordance with their anticipated roles and responsibilities.	1.73 ± 0.63	2 ± 0	1.25 ± 0.89	2 ± 0	0.019 *	1.63 ± 0.72	2 ± 0	0.223	1.56 ± 0.88	1.85 ± 0.38	0.299
Identify domestic support measures that could enhance staff flexibility for shift work and longer working hours.	1.91 ± 0.43	2 ± 0	1.75 ± 0.71	2 ± 0	0.438	1.88 ± 0.5	2 ± 0	0.553	2 ± 0	1.85 ± 0.56	0.419
Ensure the availability of the services of multidisciplinary psychosocial support teams.	0.82 ± 0.96	1.18 ± 0.98	0.63 ± 0.92	0 ± 0	0.127	0.44 ± 0.81	1.83 ± 0.41	0.001 **	0.89 ± 0.93	0.77 ± 1.01	0.781
Use occupational health mechanisms that ensure the wellbeing and safety of personnel.	1.14 ± 0.99	1.45 ± 0.93	0.63 ± 0.92	1.33 ± 1.16	0.188	0.81 ± 0.98	2 ± 0	0.008 **	1.22 ± 0.97	1.08 ± 1.04	0.744
Establish a policy to monitor and manage staff suspected of or confirmed as having COVID-19 or who have been exposed to COVID-19 patient.	1.91 ± 0.43	2 ± 0	1.75 ± 0.71	2 ± 0	0.438	1.88 ± 0.5	2 ± 0	0.553	2 ± 0	1.85 ± 0.56	0.419
Train relevant health workers on screening and triage, clinical case management and infection control.	1.77 ± 0.53	1.82 ± 0.41	1.63 ± 0.74	2 ± 0	0.555	1.75 ± 0.58	1.83 ± 0.41	0.751	1.89 ± 0.33	1.69 ± 0.63	0.404

* Level of significance *p* < 0.05, ** level of significance *p* < 0.01.

**Table 6 ijerph-20-01798-t006:** Detailed average compliance score of participating hospitals according to response function stratified according to hospital location, type, and years of accreditation for response functions (Surge Capacity, Continuity of Essential Services, Rapid Identification, and Diagnosis).

Readiness Response	Overall	Hospital Location				Hospital Type			Years of Accreditation		
	Total Score (*n* = 22)	Middle (*n* = 11)	North (*n* = 8)	South (*n* = 3)	*p*-Value	Public (16)	Private (6)	*p*-Value	<5 Years (*n* = 9)	≥5 Years (*n* = 13)	*p*-Value
**Surge Capacity**											
Identify ways of expanding hospital in-patient capacity including physical space, staff, supplies and processes.	1.55 ± 0.86	1.64 ± 0.81	1.25 ± 1.03	2 ± 0	0.403	1.38 ± 0.96	2 ± 0	0.131	1.78 ± 0.67	1.38 ± 0.96	0.302
Calculate maximum case admission capacity	1.64 ± 0.73	1.73 ± 0.65	1.38 ± 0.92	2 ± 0	0.395	1.5 ± 0.82	2 ± 0	0.155	1.67 ± 0.71	1.62 ± 0.77	0.875
Estimate the maximum number of patients’ rooms that can be converted into isolation rooms and the maximum number of patients to be placed in isolation rooms.	1.73 ± 0.63	1.91 ± 0.30	1.38 ± 0.92	2 ± 0	0.137	1.63 ± 0.72	2 ± 0	0.223	1.78 ± 0.44	1.69 ± 0.75	0.763
Coordinate with the hospital administrative board and local authorities to services other than the existing hospital facilities.	1.91 ± 0.43	2 ± 0	1.75 ± 0.71	2 ± 0	0.438	1.88 ± 0.5	2 ± 0	0.553	2 ± 0	1.85 ± 0.55	0.419
Adapt admission and discharge criteria and prioritize patients and clinical interventions according to available treatment capacity and demand.	1.82 ± 0.59	1.82 ± 0.60	1.75 ± 0.71	2 ± 0	0.835	1.88 ± 0.5	1.67 ± 0.82	0.473	1.56 ± 0.88	2 ± 0	0.081
**Continuity of Essential Services**											
List all hospital services in priority order and identify nonessential services that could be suspended if necessary.	1.68 ± 0.72	1.91 ± 0.30	1.25 ± 1.04	2 ± 0	0.095	1.63 ± 0.81	1.83 ± 0.41	0.556	2 ± 0	1.46 ± 0.88	0.083
Identify resources needed to ensure continuity of those hospital services identified as essential.	1.73 ± 0.63	1.82 ± 0.60	1.5 ± 0.76	2 ± 0	0.421	1.63 ± 0.72	2 ± 0	0.223	1.78 ± 0.67	1.69 ± 0.63	0.763
Determine strategies to maintain services for at-risk patients during the outbreak period.	1.77 ± 0.61	1.82 ± 0.60	1.63 ± 0.74	2 ± 0	0.647	1.69 ± 0.70	2 ± 0	0.297	2 ± 0	1.62 ± 0.77	0.152
**Rapid Identification**											
Train health workers for accurate rapid identification and timely reporting of suspected cases.	1.95 ± 0.21	2 ± 0	1.88 ± 0.35	2 ± 0	0.438	1.94 ± 0.25	2 ± 0	0.553	2 ± 0	1.92 ± 0.28	0.419
Have a triage procedure in place in the emergency department with a well-equipped triage station at the entrance of the facility, supported by trained staff.	1.68 ± 0.65	1.55 ± 0.82	1.88 ± 0.35	1.67 ± 0.58	0.57	1.56 ± 0.73	2 ± 0	0.162	1.78 ± 0.67	1.62 ± 0.65	0.575
Develop a system for alternative triage, for example, a telephone triage.	1.09 ± 1.02	0.91 ± 1.04	1.25 ± 1.04	1.33 ± 1.16	0.72	0.88 ± 1.03	1.67 ± 0.82	0.106	0.89 ± 1.05	1.23 ± 1.01	0.453
**Diagnosis**											
Ensure the continuous availability of laboratory and imaging services for diagnosis of COVID-19.	2 ± 0	2 ± 0	2 ± 0	2 ± 0		2 ± 0	2 ± 0		2 ± 0	2 ± 0	
Develop procedures and train staff in taking samples and in properly handling, packaging, and transporting to the designated laboratory.	2 ± 0	2 ± 0	2 ± 0	2 ± 0		2 ± 0	2 ± 0		2 ± 0	2 ± 0	
Ensure mechanisms for the prompt provision of laboratory data to the physicians, front-line workers, and health authorities.	1.91 ± 0.43	2 ± 0	1.75 ± 0.71	2 ± 0	0.438	1.88 ± 0.5	2 ± 0	0.553	1.78 ± 0.67	2 ± 0	0.238
Establish a laboratory referral pathway for the identification, confirmation, and monitoring of COVID-19.	1.91 ± 0.43	2 ± 0	1.75 ± 0.71	2 ± 0	0.438	1.88 ± 0.5	2 ± 0	0.553	1.78 ± 0.67	2 ± 0	0.238

**Table 7 ijerph-20-01798-t007:** Detailed average compliance score of participating hospitals according to response function stratified according to hospital location, type, and years of accreditation for response functions (isolation and case management, and infection prevention and control).

Readiness Response	Overall	Hospital Location				Hospital Type			Years of Accreditation		
	Total Score (*n* = 22)	Middle (*n* = 11)	North (*n* = 8)	South (*n* = 3)	*p*-Value	Public (16)	Private (6)	*p*-Value	<5 Years (*n* = 9)	≥5 Years (*n* = 13)	*p*-Value
**Isolation and Case Management**											
Develop and implement hospital strategy for the admission, referral, internal transfer and discharge of patients.	1.91 ± 0.43	2 ± 0	1.75 ± 0.71	2 ± 0	0.438	1.88 ± 0.5	2 ± 0	0.553	1.78 ± 0.67	2 ± 0	0.238
Identify, add signage, and equip areas for the medical care of suspected and confirmed cases in secure and isolated conditions.	1.77 ± 0.61	1.64 ± 0.81	1.88 ± 0.35	2 ± 0	0.576	1.81 ± 0.54	1.67 ± 0.82	0.63	1.78 ± 0.67	1.77 ± 0.6	0.975
Patients should be placed in an adequately ventilated single room (≥12 air changes/hour). When single rooms are not available, patients suspected of having COVID-19 should be grouped together.	1.27 ± 0.55	1.36 ± 0.51	1.13 ± 0.64	1.33 ± 0.58	0.656	1.13 ± 0.5	1.67 ± 0.52	0.036 *	1.33 ± 0.5	1.23 ± 0.6	0.678
Provide guidelines/protocols for the management of suspected or confirmed cases.	1.95 ± 0.21	1.91 ± 0.30	2 ± 0	2 ± 0	0.629	2 ± 0	1.83 ± 0.41	0.104	1.89 ± 0.33	2 ± 0	0.238
Team of adequately trained healthcare workers should be designated to care exclusively for suspected or confirmed cases.	2 ± 0	2 ± 0	2 ± 0	2 ± 0		2 ± 0	2 ± 0		2 ± 0	2 ± 0	
Maintain a record of all people entering each patient’s room, including all staff and visitors.	1.32 ± 0.89	1.36 ± 0.92	1.13 ± 0.99	1.67 ± 0.58	0.673	1.19 ± 0.91	1.67 ± 0.82	0.273	1.67 ± 0.71	1.08 ± 0.95	0.131
Avoid moving and transporting any patient out of their room or area unless it is medically necessary.	1.91 ± 0.29	2 ± 0	1.75 ± 0.46	2 ± 0	0.161	1.88 ± 0.34	2 ± 0	0.388	2 ± 0	1.85 ± 0.38	0.237
**Infection Prevention and Control**											
Ensure that HCPs, patients, and visitors are aware of respiratory and hand hygiene and the prevention of healthcare-associated infections.	1.77 ± 0.43	1.82 ± 0.41	1.75 ± 0.46	1.67 ± 0.58	0.861	1.75 ± 0.45	1.83 ± 0.41	0.695	1.89 ± 0.33	1.69 ± 0.48	0.302
Ensure availability and proper use of protective supplies according to risk stages of clinical posts.	1.95 ± 0.21	2 ± 0	1.88 ± 0.35	2 ± 0	0.438	1.94 ± 0.25	2 ± 0	0.553	2 ± 0	1.92 ± 0.28	0.419
Limit visitors to those essential for patient support. Ensure that visitors apply droplet and contact precautions.	2 ± 0	2 ± 0	2 ± 0	2 ± 0		2 ± 0	2 ± 0		2 ± 0	2 ± 0	
Ensure the facility has infrastructure and procedures for proper hand hygiene, including hand washing, continuous training and supplies.	1.95 ± 0.21	2 ± 0	1.88 ± 0.35	2 ± 0	0.438	1.94 ± 0.25	2 ± 0	0.553	2 ± 0	1.92 ± 0.28	0.419
Have protocols or procedures available for cleaning and hygiene of clinical areas.	1.91 ± 0.29	2 ± 0	1.88 ± 0.35	1.67 ± 0.58	0.209	1.88 ± 0.34	2 ± 0	0.388	2 ± 0	1.85 ± 0.38	0.237
Ensure the health facility has dedicated area(s) and protocols for the disinfection and sterilization of biomedical equipment and material devices.	2 ± 0	2 ± 0	2 ± 0	2 ± 0		2 ± 0	2 ± 0		2 ± 0	2 ± 0	
Ensure the healthcare facility has a protocol and a marked route for the management and final disposal of infectious biological waste.	1.86 ± 0.47	1.82 ± 0.60	1.88 ± 0.35	2 ± 0	0.847	1.94 ± 0.25	1.67 ± 0.82	0.235	2 ± 0	1.77 ± 0.6	0.265
Ensure strict supervision on the implementation of infection prevention and control measures.	1.55 ± 0.74	1.82 ± 0.41	1.13 ± 0.99	1.67 ± 0.58	0.122	1.5 ± 0.82	1.67 ± 0.52	0.649	2 ± 0	1.23 ± 0.83	0.012 *

* Level of significance *p* < 0.05.

**Table 8 ijerph-20-01798-t008:** Predictors of Continuity of Essential Services (dependent variable).

Coefficients					
Independent Variables	Unstandardized Coefficients		Standardized Coefficients	t	Sig.
	B	Std. Error	Beta		
(Constant)	−1.895	0.524		−3.618	0.002 **
Operational Support	0.947	0.235	0.403	4.025	0.001 **
Human Resources	1.000	0.162	0.851	6.164	0.000 **
Surge Capacity	−0.026	0.118	−0.031	−0.224	0.826

** level of significance *p* < 0.01, R2: 0.820 (Predictors: (Constant), Surge Capacity, Operational Support, Human Resources).

## Data Availability

Data sharing is possible from corresponding authors upon request.

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
