# Peer review of "Assessment of Hospital Readiness to Respond to COVID-19 Pandemic in Jordan—A Cross Sectional Study"

_ijerph, 2023, doi:10.3390/ijerph20031798_

Round 1
Reviewer 1 Report
The manuscript is generally well-structured. The authors performed a cross-sectional analysis of 22 hospitals in Jordan to evaluate the hospital readiness to covid-19 pandemic.
Major Comments:
1. For 2.6 Statistical data analysis, which specific independent sample test was used for the two-sample analysis?
2. The sentence spacing should be consistent. For example, line 154, line 211, line 224 and line 228 seem to have larger sentence spacing.
3. Table 4 should be complete.
4. For 3.4 Regression analysis, in terms of predictability, which criteria were you using, whether the predictors are statistically significant? It would be better to attach the R2 value if you have built the regression models. This will reflect the correlation between the observed value and the predicted value.
Reviewer 2 Report
Dahmash et al. carried a cross sectional study to assess the level of readiness of hospitals in Jordan to respond to the COVID-19 pandemic using the WHO hospital readiness checklist. The results identified gaps in practices particularly the planning and provision of psychological support for staff wellbeing. The findings will assist hospital leaders in making informed decisions, developing a roadmap, and creating strategies to respond to COVID-19 and other emergency cases in their hospitals. Overall, I think this is a good article for the audience of the journal. The article requires minor revisions to be suitable for publication. I have the following comments:
1- Why only accredited hospitals were enrolled in this study.
2- What was the role of HCAC in tool validation
3- In methods section 2.4: the survey process: how was conflict of interest addressed (elaborate).
4- What was the total number of surveyors enrolled in this research. How did HCAC avoid bias in data collection.
5- Revise table 8 for numbers.
6- Elaborate on leadership role in pandemic and how was that scored low in your discussion.
7- Why private hospitals were only from the middle central region of the country
8- Why postmortem plans was affected particularly in public hospitals
9- Continuity of essential services was a key challenge worldwide. Why the northern region showed lowest compliance in this domain.
